

# Distilroberta2gnn: a new hybrid deep learning approach for aspect-based sentiment analysis

Aseel Alhadlaq* and Alaa Altheneyan*

Department of Computer Science and Engineering, College of Applied Studies and Community Service, King Saud University, Riyadh, Saudi Arabia
* These authors contributed equally to this work.

## ABSTRACT

In the field of natural language processing (NLP), aspect-based sentiment analysis (ABSA) is crucial for extracting insights from complex human sentiments towards specific text aspects. Despite significant progress, the field still faces challenges such as accurately interpreting subtle language nuances and the scarcity of high-quality, domain-specific annotated datasets. This study introduces the Distil-RoBERTa2GNN model, an innovative hybrid approach that combines the DistilRoBERTa pre-trained model's feature extraction capabilities with the dynamic sentiment classification abilities of graph neural networks (GNN). Our comprehensive, four-phase data preprocessing strategy is designed to enrich model training with domain-specific, high-quality data. In this study, we analyze four publicly available benchmark datasets: Rest14, Rest15, Rest16-EN, and Rest16-ESP, to rigorously evaluate the effectiveness of our novel DistilRoBERTa2GNN model in ABSA. For the Rest14 dataset, our model achieved an F1 score of 77.98%, precision of 78.12%, and recall of 79.41%. The Rest15 dataset shows that our model achieves an F1 score of 76.86%, precision of 80.70%, and recall of 79.37%. For the Rest16-EN dataset, our model reached an F1 score of 84.96%, precision of 82.77%, and recall of 87.28%. For Rest16-ESP (Spanish dataset), our model achieved an F1 score of 74.87%, with a precision of 73.11% and a recall of 76.80%. These metrics highlight our model's competitive edge over different baseline models used in ABSA studies. This study addresses critical ABSA challenges and sets a new benchmark for sentiment analysis research, guiding future efforts toward enhancing model adaptability and performance across diverse datasets.

# INTRODUCTION

Analyzing public sentiment has become increasingly crucial in light of the vast amount of information available on the internet. Social media platforms have facilitated the transparent expression of opinions on various subjects, including companies, products, services, events, news, and individuals. Analysing users' opinions is valuable; however, the structure and volume of these opinions limit the extraction and analysis process. The challenges in sentiment analysis (SA), a branch of natural language processing (NLP) that

Corresponding author
Aseel Alhadlaq,
asalhadlaq@ksu.edu.sa

identifies sentiments in user reviews, have spurred researchers' interest and dedication to the field. SA encompasses a broad spectrum of applications across numerous domains, enabling business leaders to extract valuable insights from client feedback to improve product quality and customer service and extend its utility to fields such as politics, film criticism, social media monitoring, market research, and healthcare. This demonstrates its significant impact and versatility in analyzing and interpreting sentiments from various textual data sources (*Yadav & Vishwakarma, 2020*).

Sentiment analysis (SA) distinguishes sentiments at document, sentence, and aspect levels, enabling a nuanced understanding of textual feedback. However, these approaches may not capture the complexity of mixed sentiments within a single statement, necessitating aspect-based SA for a more granular analysis. For instance, the multifaceted review: "The steak was perfectly cooked and seasoned to my liking, the noise level in the dining area was uncomfortably loud, yet the waitstaff's attentiveness was just average." Here, a positive sentiment is attributed to the "food" aspect, reflecting satisfaction with the steak's preparation. Conversely, the "ambiance" aspect receives a negative sentiment due to the critique of the loud dining area, indicating discomfort. Meanwhile, the "service" aspect is met with a neutral sentiment, implying that the service was neither notably good nor bad but rather average. This example underscores the importance of aspect-based SA in dissecting complex feedback, where distinct sentiments are expressed towards various elements of an experience, providing a comprehensive and detailed sentiment analysis.

These days, ABSA is an essential tool in sentiment analysis (SA), crucial for understanding the subtle nuances of individual opinions. This technique identifies detailed sentiments towards specific aspects of products or services, enabling companies and researchers to distinguish areas worthy of praise accurately or requiring improvement. By utilizing this depth of analysis, businesses can adjust their strategies to meet their customers' intricate preferences and needs more effectively. Thus, ABSA plays a key role in transforming complex sentiment data into actionable insights, facilitating a more precise and impactful approach to enhancing customer satisfaction and product innovation.

Expanding upon the discourse surrounding the essential role of ABSA in sentiment analysis, the scholarly literature delineates three principal methodologies: machine learning (ML), lexicon-based, and hybrid approaches (*Thakkar & Patel, 2015*). Predominantly, research in this domain has harnessed supervised learning algorithms, notably support vector machines (SVM) and Naive Bayes (NB), for the formulation of classification models (*Sankar & Subramaniyaswamy, 2017*). In contrast, the lexicon-based strategy entails the compilation of an extensive lexicon that associates sentiment-laden terms with their respective polarity scores. This strategy bifurcates into dictionary-based lexicons, devised for broad applicability, and *corpus*-based lexicons, meticulously curated for specific thematic areas. Although ontology-based lexicons constitute an alternative strategy, their adoption has been comparatively limited, attributed to the arduous and time-intensive endeavours required to develop and apply an ontology-based *corpus* (*Bandari & Bulusu, 2020*). The hybrid model seeks to amalgamate the predictive prowess of ML techniques with the nuanced understanding afforded by lexicon-based methods, culminating in a more robust analytical framework for sentiment analysis. This confluence

of methodologies epitomizes the interdisciplinary and innovative essence of sentiment analysis research, continually striving to refine and augment the efficacy of sentiment detection across diverse discursive landscapes.

The primary aim of ABSA is to identify relationships between aspects and their corresponding sentiment-laden words, thereby determining the sentiment polarity for each aspect. Traditional ABSA methods often rely on sentiment dictionaries to allocate sentiment polarities or scores to words expressing opinions (*Zhang et al., 2018*). These scores are then integrated, considering their weights, to ascertain the general sentiment direction. However, this approach requires the creation and maintenance of sentiment dictionaries, posing considerable challenges. Some methods of analyzing aspect sentiment that depends on machine learning involve using handcrafted features to educate classifiers, necessitating extensive manual effort (*Mubarok, Adiwijaya & Aldhi, 2017*). The advent of deep learning has sparked considerable interest among researchers and has found applications across various domains. Earlier studies have employed recurrent neural network (RNN) models (*Dong et al., 2014*) to process the sequential nature of sentences and unearth semantic connections between aspects and sentiment words. However, these models often struggle with issues like semantic information loss and gradient vanishing. To address these limitations, long short-term memory (LSTM) networks (*Kuppusamy & Selvaraj, 2023*) have been widely adopted in ABSA tasks, showcasing their ability to manage long sequences and capture prolonged dependencies. Recently, the focus has shifted towards leveraging graph neural networks, specifically graph convolutional networks (GCN) and graph attention networks (GAT), for sentiment analysis due to their significant progress in the field.

Researchers have used pre-trained language models (PLM) to make unified frameworks for the ABSA problem, achieving impressive performance on benchmark datasets (*Chen et al., 2021*; *Yan et al., 2021*). A fundamental difficulty identified in PLM fine-tuning is the considerable difference in objective forms between the pre-training and fine-tuning phases, which stops PLMs from reaching their full potential. Prompt-tuning, a new approach introduced by recent studies (*Jiang et al., 2020*; *Shin et al., 2020*; *Schick & Schütze, 2020*; *Han et al., 2022*), aims to align the objective functions of pre-training and fine-tuning. Using suitable prompts and tuning targets, this method can fine-tune the model's behaviour to suit various downstream tasks better, making PLMs better at handling specific information and possibly improving model performance for ABSA. However, making an effective prompt for ABSA requires domain expertise, and its automatic generation may increase computational costs for validation. Additionally, the use of advanced models like BERT (*Zhu et al., 2023*) and the integration of graph neural networks (GNNs) (*Liu & Zhou, 2022*) have significantly improved the understanding of sentiment at the aspect level, enriching the field of ABSA with the capability to understand complex user sentiments and analyze relational data with remarkable precision. *Chen, Fan & Wang (2024)* introduced the SS-GCN model, which innovatively incorporated both syntactic and semantic information into graph neural networks for detailed sentiment analysis of specific aspects. This model distinguished itself by autonomously learning a matrix that weights syntactic connections and utilizing semantic interrelations to enrich

textual representations. By emphasizing the significance of syntactic connections and incorporating semantic contexts that are frequently overlooked by similar models, the SS-GCN offered a deeper, more refined comprehension of textual content. *Jiang, Xu & Liu (2023)* introduced the MLGCN (merge aspect entities and location-aware transformation graph convolutional network) model, designed to enhance aspect-level sentiment analysis significantly. The MLGCN model features two distinct variants, each tailored to address different syntactic complexities within text data, thereby facilitating a more nuanced approach to sentiment analysis. This innovative model aims to bridge gaps in current sentiment analysis techniques by integrating advanced syntactic and semantic processing.

This combination of advancements shows sentiment analysis research's changing and improving nature, highlighting the continuous search for more insightful analytical methods. Nonetheless, the search for efficiency and accuracy in sentiment analysis, especially aspect-based approaches, needs more research. Despite progress with PLMs, BERT, and GNNs, the complexity of human language and sentiment expression presents ongoing challenges. Solving the gaps between pre-training objectives and fine-tuning tasks, optimizing computational resources for prompt generation, and improving the interpretability of complex models remain important areas for study. Such efforts will improve current methods and make sentiment analysis applicable to a wider range of diverse and nuanced datasets, pushing the limits of what these advanced computational models can achieve in understanding human sentiment.

This study introduces several key advancements in the field of ABSA, outlined as follows:

- In this study, we introduce the DistilRoBERTa2GNN model, a hybrid deep-learning technique explicitly designed for the complexities of ABSA. By utilizing the advanced capabilities of the DistilRoBERTa pre-trained model for precise feature extraction from textual data and synergizing it with the dynamic classification abilities of Graph Neural Networks (GNN), our approach represents a significant leap forward in the field. This strategic integration allows for a deeper and more nuanced interpretation of sentiments tied to specific textual aspects, enhancing the performance and depth of sentiment analysis.

- We employ in-depth data preprocessing steps designed to remove noisy data, which helps improve our ABSA tasks' effectiveness. Our technique involves a thorough assessment of four benchmark datasets: SemEval 2014 Task 4 Restaurant (Rest14), SemEval 2015 Task 12 Restaurant (Rest15), SemEval 2016 Task 5 Restaurant-English (Rest16-EN) and SemEval 2016 Task 5 Restaurant-Spanish (Rest16-ESP). These assessments showcase our model's resilience and its capacity to adjust to different scenarios, underscoring its potential to enhance sentiment research methodologies.

- We have extensively compared the DistilRoBERTa2GNN model with the ABSA-related studies (baseline models). This comparison highlights our model's superior performance and provides insights into the current state of ABSA methodologies. It encourages further research and innovation in the field, aiming to contribute to developing and refining sentiment analysis technologies.

# RELATED WORK

## ABSA studies

In ABSA, research has extensively employed three pivotal approaches: firstly, sentiment polarity dictionaries for intricate statistical analysis; secondly, the design of sophisticated ML frameworks; and thirdly, the innovative use of deep learning techniques, notably through hierarchical models. Sentiment polarity dictionaries, while foundational, necessitate a meticulous definition of judgment norms, thereby confining the model within the bounds of a pre-determined lexicon and limiting adaptability to nuanced sentiment expressions. Conversely, ML methods in ABSA demand comprehensive data labeling and are inherently constrained by the quality of the training dataset. To overcome these challenges, the third approach, deep learning, has been adopted, showing exceptional efficacy in discerning complex sentiment relationships related to specific aspects through models like Convolutional Neural Networks (CNNs), RNNs, and LSTMs (*Shrestha & Mahmood, 2019*).

The subsequent sections will delve deeper into ML-based ABSA, deep learning-based ABSA, and hybrid ABSA approaches, offering a comprehensive perspective on each. These discussions aim to set a foundational understanding for further exploration of each technique's nuances, advantages, and limitations within the field of ABSA.

## Machine learning (ML)-based ABSA

Traditional ML methods such as Naive Bayes (*Mubarok, Adiwijaya & Aldhi, 2017*; *Anand & Naorem, 2016*), Decision Trees/Random Forest (*Fitri, Andreswari & Hasibuan, 2019*; *Karthika, Murugeswari & Manoranjithem, 2019*), and Support Vector Machines (SVM) (*Pannala et al., 2016*; *Tripathy, Agrawal & Rath, 2016*) have been extensively utilized. *Esuli, Moreo & Sebastiani (2020)* introduced a cross-lingual sentiment analysis technique utilizing structural correspondence learning (SCL). This method enables knowledge transfer by establishing a correspondence between pivot terms in two different feature spaces, and it can be adapted to various classifiers. With the rapid proliferation of deep learning in every aspect of computer science, a multitude of deep learning strategies are now being adopted for identifying sentiments.

## Deep learning-based ABSA

*Chen (2015)* implemented convolution and max pooling operations to identify key local features for sentiment analysis. Nonetheless, this technique faces challenges in the context of ABSA due to sentences containing multiple aspects, which CNNs might not distinguish accurately, thus affecting its efficacy. In response, *Xue & Li (2018)* developed a gated convolutional network with aspect embedding (GCAE) designed to delineate the interplay between aspects and their contexts. This model utilizes a CNN for the extraction of n-gram features from embeddings, while a gating mechanism is employed to formulate the final sentiment analysis. Moreover, the practice of averaging aspect embeddings within CNNs can obscure the meaning of irrelevant words and omit sequence information pertinent to a specific aspect (*Liu & Shen, 2020*). Addressing this issue, *Li et al. (2018)* introduced a

target-specific transformation mechanism, enabling the generation of word representations that are specific to the target aspect.

*Mohammad et al. (2023)* suggested a deep learning model that relied on features retrieved from GRU using the Multilingual Universal Sentence Encoder (MUSE). The recommended "pooled-GRU model" incorporates two primary objectives of ABSA: aspect polarity classification and aspect extraction. The suggested method has achieved satisfactory results within a specific range. The results of the suggested model have shown good performance compared to the baseline model, and the related research has confirmed the accuracy of the same datasets. Furthermore, the suggested model has improved the F1 score.

*Onan (2022)* conducted a study on a Bidirectional-Convolutional Recurrent Neural Network (B-CRNN) for sentiment analysis. The experimental conditions have surpassed the B-CRNN's performance when integrated with current methods. *Xu et al. (2022)* developed a Social Relations-Guided Multi-attention Networks (SRGMANs) model to analyze visual sentiment by analyzing several images. The examination of the results indicates that the proposed model has demonstrated superior performance in analyzing the sentiment of social photos.

## Hybrid ABSA

Numerous research works have proposed combining the strengths of lexicon-based methods with machine learning (ML) techniques, especially deep learning, for enhanced ABSA performance. *Meškele & Frasincar (2020)* introduced an approach that integrates a lexicalized domain ontology with an attention-based neural model for ABSA tasks. Similarly, *Asif et al. (2020)* developed a hybrid framework for analyzing sentiment in social media texts, employing a domain-specific multilingual lexicon for identifying posts and comments with extremist content alongside a machine-based technique for sentiment classification on social media platforms. This hybrid approach significantly reduces the labor and time of annotating large text corpora for classifier training purposes.

*Ayetiran (2022)* suggested a CNN-BiLSTM method for combined attention. It extracts symbolic characters and contextual text representations by learning emotion statistics at the document level. Attention-based models fail to consider the impact of sentence patterns and grammatical dependence information. Failure to use this extra semantic information could result in inaccurate outcomes when determining the sentiment orientation towards specific aspects. In the study by *Chen, Fan & Wang (2024)*, they presented the SS-GCN model. This novel approach integrated both syntactic and semantic data into graph neural networks for detailed sentiment analysis concerning specific aspects. This model was distinguished by its capability to autonomously learn a matrix that weights syntactic connections and utilizes semantic interrelations to enrich textual representations. By giving due importance to syntactic connections and incorporating semantic contexts that are frequently overlooked by similar models, the SS-GCN offered a deeper, more refined comprehension of textual content. Comprehensive experimental results showed that SS-GCN outperformed established baseline models, with particularly impressive F1 scores of 74.26% on the Rest14 dataset, 71.78% on Lap14, 66.16% on Rest15,

and 74.28% on Rest16-EN, demonstrating its effectiveness in capturing and analyzing sentiment at a granular level. In the study of *Jiang, Xu & Liu (2023)*, they developed the MLGCN (Merge aspect entities and Location-aware transformation graph convolutional network), a sophisticated graph convolutional network aimed at refining aspect-level sentiment analysis. In their study, two variants were utilised (MLGCN-DL and MLGCN-SE). The MLGCN-DL variant enhances at utilising syntactic dependencies, which is evident from its impressive performance, achieving F1 scores of 74.17% on LAP14 and 77.61% on REST14. On the other hand, the MLGCN-SE variant is designed to manage more complex syntactic structures through a dynamic Squeeze-and-Excitation attention mechanism, reflecting its adaptability with F1 scores of 74.40% on Twitter and 76.05% on REST16-EN. These results highlight the model's ability to enhance sentiment analysis across diverse textual contexts by incorporating advanced syntactic and semantic insights, setting a new benchmark in the field and providing a critical evaluation of existing methodologies. The EK-GCN model, developed by *Gu et al. (2023)*, significantly advances aspect-based sentiment analysis by integrating external knowledge, such as sentiment lexicons and part-of-speech information, into a graph convolutional neural network framework. This innovative approach addresses the limitations of previous models by more effectively capturing intricate word relationships and the nuances of edge labels. Demonstrated across four benchmark datasets, the EK-GCN model shows substantial improvements in F1 scores, achieving 74.93% on REST14 and 76.54% on LAP14, highlighting its enhanced capability to analyze complex sentiment expressions.

## METHODOLOGY

In this study, an ABSA and detection is proposed. For this purpose, we use four publicly available benchmark datasets. After the data collection, a sequential four-phase data preprocessing protocol is applied to the datasets. The preprocessing commences with data cleaning to purge irrelevant or erroneous data, establishing a clean, reliable foundation. Subsequently, the data undergoes a phase of emphasis, where significant features are highlighted to bolster the model's ability to discern nuanced sentiments related to specific aspects within the text. Further, the data is tokenized, converting the text into a format that neural networks can efficiently process. The final phase involves constructing the data into a TensorFlow-compliant format, laying the groundwork for seamless model training and evaluation. Then, the processed training data is utilized to train the deep learning model. Finally, the testing data is used to check the performance and efficiency of our proposed model. Figure 1 demonstrates the workflow of our proposed approach.

### Data description

For this experiment, we utilize four publicly available benchmark datasets: (1) SemEval 2014 task 4 Restaurant (Rest14) (*Kirange, Deshmukh & Kirange, 2014*), (2) SemEval 2015 task 12 Restaurant (Rest15) (*Papageorgiou et al., 2015*), (3) SemEval 2016 task 5 Restaurant English (Rest16-EN) (*Pontiki et al., 2016*), and (4) SemEval 2016 task 5 Restaurant Spanish (Rest16-ESP) (*Pontiki et al., 2016*). Every dataset is divided into training and testing subsets, wherein each review sentence includes one or several aspects along with their
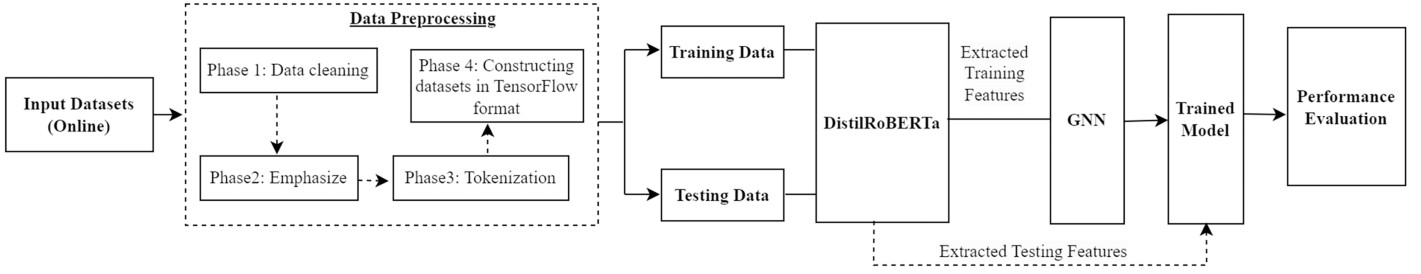

**Figure 1 Workflow of our proposed approach.**

respective sentiment orientations, which are classified as positive, negative, or neutral.
Table 1 provides detailed representations of the datasets for reference.

### SemEval-2014 Task 4 Rest14

This dataset is derived from the SemEval 2014 challenge. It features an extensive
compilation of restaurant reviews, methodically divided into training and testing subsets.
The unique aspect of this dataset lies in its detailed annotation of sentences with one or
more specific aspects of the restaurant experience, such as food quality, ambience, or
service. Each aspect is evaluated for sentiment polarity, with annotations distinguishing
between positive, neutral, or negative sentiments. This granularity allows for a nuanced
analysis of customer opinions and preferences. This dataset provided a robust foundation
for our analysis, with the training set comprising 2,164 positives, 633 neutral, and 805
negative annotations, totalling 3,602 instances. The testing set included 728 positive, 196
neutral, and 196 negative instances, culminating in 1,120 entries. This diverse assortment
of sentiments offered a rich canvas for our sentiment analysis exploration.

### SemEval-2015 Task 12 Rest15

Building on the foundations laid by Rest14, the Rest15 dataset from SemEval 2015 expands
the horizon of sentiment analysis within the restaurant sector. It adheres to a similar
structure, with its content organized into training and testing sets. Sentences within the
reviews are tagged with aspects pertaining to the dining experience, accompanied by their
respective sentiment polarities. The categorization into positive, neutral, and negative
sentiments provides an invaluable resource for dissecting the multifaceted nature of
customer feedback. For Rest15, the training set yielded 912 positive, 36 neutral, and 256
negative sentiments among 1,204 total instances. The testing dataset comprised 326
positive, 34 neutral, and 182 negative sentiments, aggregating 542 instances. The relatively
smaller volume of neutral sentiments presented unique challenges and insights into
sentiment distribution within this dataset.

### SemEval-2016 Task 5 Rest16-EN

The Rest16-EN dataset continues the sequence emanating from SemEval 2016's Task 5. It
encapsulates a rich dataset of restaurant reviews, again segmented into training and testing

**Table 1 Four datasets with partition.**

| Dataset | Training | | | | Testing | | | |
|---|---|---|---|---|---|---|---|---|
| | Positive | Neutral | Negative | Total | Positive | Neutral | Negative | Total |
| Rest 14 | 2,164 | 633 | 805 | 3,602 | 728 | 196 | 196 | 1,120 |
| Rest 15 | 912 | 36 | 256 | 1,204 | 326 | 34 | 182 | 542 |
| Rest 16-EN | 1,240 | 69 | 439 | 1,748 | 468 | 30 | 117 | 615 |
| Rest 16-ESP | 1,925 | 120 | 674 | 2,719 | 750 | 48 | 274 | 1,072 |

sets, with each sentence annotated for specific aspects and their sentiment polarities. This dataset not only carries forward the tradition of its predecessors in providing a detailed look at the spectrum of customer sentiment but also introduces new challenges and opportunities for deeper sentiment analysis through its nuanced annotations. The dataset from Rest16-EN contributed 1,240 positive, 69 neutral, and 439 negative sentiments within the training set, amounting to 1,748 instances. The testing set accounted for 468 positive, 30 neutral, and 117 negative sentiments, leading to 615 instances. The nuanced distribution of sentiments across this dataset provided valuable perspectives on customer feedback trends.

### SemEval-2016 Task 5 Rest16-ESP

The dataset Rest16-ESP is designed for Aspect-Based Sentiment Analysis in the restaurant domain. It contains annotated reviews in Spanish, where each review includes various aspects such as food, service, ambience, and price, each tagged with a sentiment that can be positive, negative, or neutral. The dataset aims to identify specific terms or phrases in the reviews that indicate discussed aspects and determine the sentiment expressed towards them, considering the context of the entire review or sentence. This dataset is valuable for developing and evaluating models automatically identifying and classifying aspects and their sentiments within the text, advancing sentiment analysis techniques for Spanish-language content in the restaurant domain. The dataset includes 1,925 positive, 120 neutral, and 674 negative sentiments within the training set, totalling 2,719 instances, and 750 positives, 48 neutral, and 274 negative sentiments in the testing set, totalling 1,072 instances.

### Data preprocessing techniques

The datasets collected from the sources were in XML format. Here, we provide an example of XML snippet of our datasets, which consists of a `sentence` element identified by an `id`, encapsulating a `text` node that contains a specific statement. Nested within the sentence, there's an `Opinions` container holding multiple `Opinion` elements. Each `Opinion` element is attributed with details such as `target`, reflecting the focus of the sentiment; `category`, indicating the aspect category; `polarity`, denoting the sentiment's nature; and `from` and `to` attributes, specifying the character range of the target within the text.

```
<sentence id = "1119578:2">
<text> Yes, they use fancy ingredients, but even fancy
     ingredients don't make for good pizza unless someone knows
     how to get the crust right.</text>
<Opinions>
  <Opinion target="ingredients" category="FOOD#QUALITY"
     polarity="positive" from="20" to="31"/>
  <Opinion target="pizza" category="FOOD#QUALITY" polarity="negative"
     from="80" to="85"/>
  <Opinion target="crust" category="FOOD#QUALITY" polarity="negative"
     from="122" to="127"/>
</Opinions>
</sentence>
```

In our study, after an initial review of datasets, we streamlined the dataset by removing redundant information. This process refined our analysis, focusing on the most impactful data, and facilitated a more efficient examination of sentiment trends within the textual feedback. Table 2 outlines each sentence along with associated aspects and their corresponding sentiment polarities.

### Phase-1: text preprocessing techniques

In our approach to ABSA, we begin by meticulously preprocessing the text data to ensure it is primed for identifying and evaluating sentiments towards specific aspects within the text. Our first step involves the removal of HTML tags using a parsing library. This is essential for isolating the raw text from any HTML formatting, allowing us to focus exclusively on the textual content without distraction from web markup, which is extraneous to our sentiment analysis. As we progress through the preprocessing stages, our efforts are directed towards further refining the text to enhance its clarity and uniformity, thereby facilitating a more accurate sentiment analysis. We remove content enclosed within square brackets to eliminate any metadata, annotations, or extraneous information that does not contribute to the main sentiment or aspect being analyzed. Additionally, we meticulously cleanse the text of URLs, usernames, and other patterns that might introduce noise into our dataset, recognizing that such elements often detract from the meaningful sentiment analysis of specific aspects. Our preprocessing also includes removing punctuation and non-alphabetical characters, simplifying the text to its essential elements. This step is crucial for avoiding potential confusion caused by symbols or numbers, allowing us to focus on the words and their sentiments towards aspects. By converting all text to lowercase, we ensure uniformity, treating variations of a word, regardless of capitalization, as identical. This uniformity is vital for maintaining consistency in our analysis and enabling a more accurate evaluation of sentiments directed at specific features or attributes within the text.

**Table 2 Dataset after initial processing.**

| Sentence | Aspect | Polarity |
|---|---|---|
| Yes, they use fancy ingredients, but even fancy ingredients don't make for good pizza unless someone knows how to get the crust right. | Ingredients | Positive |
| Yes, they use fancy ingredients, but even fancy ingredients don't make for good pizza unless someone knows how to get the crust right. | Pizza | Negative |
| Yes, they use fancy ingredients, but even fancy ingredients don't make for good pizza unless someone knows how to get the crust right. | Crust | Negative |

These preprocessing steps are foundational to our work in ABSA. By preparing the text data carefully, we lay the groundwork for effectively discerning and assessing sentiments related to specific aspects. This preparation is not just about cleaning the data; it is about enhancing our ability to extract nuanced insights regarding how different elements or features within the text are perceived. Through this meticulous preparation, we aim to unlock deeper, more accurate understandings of sentiments within the text, thereby enriching our analysis and the insights we derive from it.

### Phase 2: emphasize preprocessing

In our pursuit of refining ABSA, we have innovated a preprocessing technique that underscores the importance of aspect terms—key phrases within sentences representing the focus of sentiment analysis. By deliberately emphasizing these aspect terms—repeating them twice and enclosing them in special tokens ([ASP] and [/ASP])—we significantly augment the model's capability to discern and assess sentiments related to specific aspects. This method, implemented before the transformer model processes the input, is designed to make aspect terms more salient, thereby influencing the attention mechanism to prioritize sentiment expressed about these aspects. The utility of this preprocessing technique lies in its ability to enhance the model's aspect awareness and improve sentiment accuracy. It leverages the transformer model's nuanced understanding of context, ensuring a sharper focus on aspect-specific sentiments. This approach facilitates more accurate sentiment analysis and contributes to the model's training and inference efficiency by guiding its attention mechanism more effectively towards the aspects of interest without significantly increasing computational complexity. Through this specialized preprocessing step, we enrich the model's input data in a particularly beneficial way for ABSA, showcasing our commitment to leveraging advancements in natural language processing for deeper, more insightful text analysis.

### Phase 3: tokenization technique

The tokenization step is undertaken following the initial preprocessing phase, where aspect terms within sentences are emphasized to enhance aspect salience. This critical stage prepares our data for the transformer-based model. A pre-trained tokenizer is employed to convert the enhanced sentences from our training and validation datasets into sequences of tokens, which serve as the fundamental input units for the transformer model. To

ensure uniformity and efficiency in processing, several critical parameters are specified during tokenization:

- **max_length:** This parameter ensures that all tokenized inputs conform to a uniform sequence length, denoted as seq_len. Sequences longer than seq_len are truncated, whereas shorter sequences are padded, standardizing input data length and facilitating batch processing.
- **truncation:** Enabled truncation ensures that inputs exceeding the predetermined maximum length (seq_len) are effectively shortened to fit within the model's architectural constraints.
- **padding:** 'Max_length' padding is applied to ensure all sequences reach the required seq_len, enhancing consistency across the dataset and enabling efficient batch processing by the model.
- **return_tensors = 'tf':** This setting indicates that the output is formatted as TensorFlow tensors, aligning with the model's framework and facilitating seamless integration into the training pipeline.

Applying these parameters during tokenization is meticulously designed to prepare the data in a manner that optimizes the transformer model's ability to learn from and interpret the emphasized aspect terms. This step is pivotal in ensuring that the nuanced semantics encoded within the enhanced sentences are effectively captured and utilized during the model training and inference phases. Through this methodological rigor in data preparation, the importance of tailored preprocessing in NLP is underscored, particularly in enhancing the efficacy of sentiment analysis models. This approach not only optimizes the model's performance but also paves the way for more sophisticated analyses in the field of ABSA.

### Phase 4: constructing datasets for TensorFlow

In our research, preparing datasets for TensorFlow-based models was a crucial step, particularly in enhancing the efficiency and effectiveness of our ML project's training and validation phases. The process is characterized by two pivotal stages: tokenization of the input texts and the structured assembly of TensorFlow datasets.

- **Tokenization:** The raw textual data was initially processed through tokenization, converting text into a machine-interpretable numerical format. This procedure yielded input_ids for the numerical representation of the texts and attention_mask to highlight the active parts of the data, optimizing the model's focus during training.
- **TensorFlow dataset assembly:** Leveraging the `tf.data.Dataset.from_tensor_slices` function, we methodically constructed the training and validation datasets. This TensorFlow utility facilitated pairing tokenized inputs with their respective labels, creating a dataset format directly consumable by TensorFlow models. This approach streamlined the data preparation workflow and ensured that our neural network architectures optimally structured the input data for processing.

Incorporating the `tf.data.Dataset.from_tensor_slices` method into our dataset preparation process was instrumental in efficiently transforming raw textual data into structured formats suitable for model training. This methodology underscores the significance of precise data handling in the ML pipeline, reflecting our commitment to fostering model accuracy and performance through meticulous data preparation. Algorithm 1 illustrates the pre-processing technique of all our phases.

## Model description

### DistilBERT

In the comprehensive study presented by *Sanh et al. (2019)*, the authors introduce DistilBERT, a model that addresses the computational inefficiencies and environmental concerns associated with the deployment of large-scale pre-trained models in NLP. Building upon the foundational work of BERT by *Devlin et al. (2018)*, DistilBERT is proposed as a distilled version that retains the core linguistic capabilities of BERT while achieving substantial reductions in size and computational demand.

At the heart of DistilBERT's development is the application of knowledge distillation, a technique where a smaller, more efficient model (the student) is trained to replicate the performance of a larger model (the teacher) Eq. (1).

$$L_{CE} = -\sum_i t_i \log(s_i) \tag{1}$$

where $t_i$ and $s_i$ denote the probability distributions predicted by the teacher and student models, respectively, for class $i$. The inclusion of this loss function allows DistilBERT to absorb nuanced probabilistic insights from BERT beyond mere label accuracy.

To further refine the distillation process, *Sanh et al. (2019)* introduce a triple loss function, which amalgamates the distillation loss $L_{CE}$, masked language modellingg loss $L_{MLM}$, and a cosine embedding loss $L_{COS}$. The latter aims to align the students' and teachers' hidden state vectors, fostering an intricate similarity in their representational spaces. This composite loss function plays a pivotal role in successfully distilling BERT's linguistic intelligence into DistilBERT, facilitating the latter's ability to achieve 97%. of BERT's performance on benchmark tasks with only 60% of its original size.

Architecturally, DistilBERT mirrors BERT but with several strategic modifications aimed at efficiency. The model eschews token-type embeddings and the pooler and reduces the number of Transformer layers by half. These design choices are underscored by an intent to minimize computational overhead without disproportionately impacting the model's performance—a empirically realised goal, as evidenced by DistilBERT's performance on the GLUE benchmark.

The practical implications of DistilBERT's design are profound. Through experimental validation, *Sanh et al. (2019)* demonstrate that DistilBERT not only maintains commendable accuracy across a suite of NLP tasks but also exhibits significantly enhanced inference speeds, particularly in on-device contexts such as mobile applications. This duality of performance and efficiency situates DistilBERT as an optimal choice for real-world applications where computational resources are at a premium.

| Algorithm 1 Pseudocode for data preprocessing. | |
| --- | --- |
| 1: **function** READ_CSV(*file_path*) | |
| 2:    **Input:** *file_path* | ▷ Path to CSV file |
| 3:    **Output:** *DataFrame* | ▷ Read CSV file and return DataFrame |
| 4:    **return** *DataFrame* | |
| 5: **end function** | |
| 6:  **function** DROP_CONFLICT_ENTRIES (*dataframe*) | |
| 7:    **Input:** *dataframe* | ▷ DataFrame to process |
| 8:    DROP_ROWS(*dataframe*[*dataframe*[*'polarity'*] == *'conflict'*].*index*) | |
| 9: **end function** | |
| 10:  **function** PREPARE_LABELS(*dataframe*) | |
| 11:    **Input:** *dataframe* | ▷ DataFrame containing polarity column |
| 12:    **Output:** *labels* | ▷ One-hot encoded labels |
| 13:    *labels* ← ONE_HOT_ENCODE(*dataframe*[*'polarity'*]) | |
| 14:    **return** *labels* | |
| 15: **end function** | |
| 16: **function** COUNT_CLASSES(*dataframe*) | |
| 17:    **Input:** *dataframe* | ▷ DataFrame to analyze |
| 18:    **Output:** *class_counts* | ▷ Counts of each class |
| 19:    *class_counts* ← *dataframe*[*'polarity'*].*value_counts*() | |
| 20:    **return** *class_counts* | |
| 21: **end function** | |
| 22: **function** PREPROCESS_TEXT(*text*) | |
| 23:    **Input:** *text* | ▷ Text to preprocess |
| 24:    **Output:** *preprocessed_text* | ▷ Preprocessed text |
| 25:    *preprocessed_text* ← REMOVE_HTML_TAGS(*text*) | |
| 26:    *preprocessed_text* ← REMOVE_SQUARE_BRACKETS_CONTENT(*text*) | |
| 27:    *preprocessed_text* ← REMOVE_URLS_AND_USERNAMES(*text*) | |
| 28:    *preprocessed_text* ← REMOVE_PUNCTUATION(*text*) | |
| 29:    *preprocessed_text* ← REMOVE_NON_ALPHABETICAL_CHARS(*text*) | |
| 30:    *preprocessed_text* ← CONVERT_TO_LOWERCASE(*text*) | |
| 31:    **return** *preprocessed_text* | |
| 32: **end function** | |

DistilBERT has been established as a robust and efficient tool for sentiment analysis across various domains, demonstrating its prowess in processing and interpreting sentiments from textual data with notable accuracy and speed. Researchers, as seen in studies by *Pramanik & Maliha (2022)*, *Xiong & Yan (2021)*, have leveraged DistilBERT for general sentiment analysis tasks, where it has shown superior performance compared to

traditional models like LSTM, especially in contexts requiring the analysis of product reviews and entity-level sentiments. These applications highlight DistilBERT's capability to balance computational efficiency and the maintenance of a high degree of linguistic understanding, making it an attractive option for analyzing sentiments in large datasets.

However, to the author's knowledge, DistilBERT has not yet been applied to the niche field of ABSA. ABSA, which requires identifying and evaluating sentiments towards specific aspects within a text, poses unique challenges that have not yet been explored with the application of DistilBERT. This gap presents an opportunity for future research to harness DistilBERT's efficient architecture and adapt it to the nuanced demands of ABSA, potentially setting a new benchmark for performance in this specialized area of sentiment analysis.

### GNN

Graphic neural networks (GNNs) represent a significant leap in the evolution of neural network architectures, enabling the effective processing of graph-structured data. This innovation traces back to the pioneering work by *Scarselli et al. (2008)*, who introduced the foundational framework for GNNs, enabling direct application of neural networks to data represented as graphs. The primary motivation behind GNNs is their capability to capture the intricate patterns within data where entities are interrelated, making them particularly suited for complex systems such as social networks, biochemical structures, and communication networks. The operational principle of GNNs revolves around the concept of message passing or neighborhood aggregation, where the feature vector of each node is updated by aggregating features from its neighbors, thereby encapsulating both local and global structural information.

The mathematical formulation (of Eq. (2)) underpinning this process is expressed through the iterative update rule:

$$h_v^{(t)} = \text{UPDATE}\left(h_v^{(t-1)}, \text{AGGREGATE}\left(\{h_u^{(t-1)} : u \in \mathcal{N}(v)\}\right)\right), \qquad (2)$$

where $h_v^{(t)}$ denotes the feature vector of node $v$ at iteration $t$, and $\mathcal{N}(v)$ represents the neighbors of $v$. This formula underscores the mechanism through which nodes integrate and update information, facilitating a deep understanding of the graph structure.

The efficacy of GNNs extends across a myriad of applications, with recent research exploring their utility in nuanced fields such as ABSA. ABSA aims to discern the sentiment towards specific aspects within a text, necessitating an understanding of both the relational aspects within the data and the contextual sentiment, a challenge aptly addressed by the relational and contextual learning capabilities of GNNs. For instance, a recent study by *Wang et al. (2022)* implemented GNNs for ABSA to identify and analyze the sentiment directed at particular aspects of product reviews, demonstrating improved accuracy over traditional models. This reflects the growing interest and potential in applying GNNs to dissect and interpret complex, relationally structured sentiment data, paving the way for more insightful and granular sentiment analysis.

The advent and evolution of GNNs underscore a pivotal advancement in ML, offering a versatile and powerful tool for analyzing graph-structured data. Their application in fields

like ABSA exemplifies their practical utility. It highlights the ongoing expansion of their applicative landscape, promising further innovations and applications in data science.

### Architecture and building procedure of our DistilRoBERTa2GNN model

In this study, we first employed the DistilRoBERTa model. The architecture of DistilRoBERTa is depicted in Fig. 2. To capture the relative positions of tokens, input embedding and positional encoding were integrated as the model's input. Multi-head attention mechanism computes the query ($Q$), key ($K$), and value ($V$) matrices by projecting the embedding dimension matrix of the sequence length. Subsequently, $Q$, $K$, and $V$ are divided by the number of heads to process scaled dot-product attention (Fig. 3) parallelly.

Scaled dot-product attention, a variant of self-attention, employs scaling by $\sqrt{dk}$ to prevent the inner product from growing excessively, where $dk$ represents the vector dimension of the $K$ matrix. The attention weights distribution, obtained by applying softmax to the dot product of $Q$ and $K$, is divided by the scaling factor $1/\sqrt{dk}$. The resulting attention value is then computed by multiplying $V$ with the attention weights distribution, as shown in Eq. (3).

$$\text{Attention}(Q, K, V) = \text{softmax}\left(\frac{QK^T}{\sqrt{dk}}\right)V \tag{3}$$

The output of scaled dot-product attention, $h_1, h_2, h_3, \ldots, h_m$, is concatenated and multiplied with a weighted matrix $W_o$, yielding the multi-head attention output matrix (Eq. (4)). Multi-head attention enables the model to attend to information at multiple locations concurrently.

$$\text{Multihead}(Q, K, V) = \text{Concat}(h_1, h_2, h_3, \ldots, h_n)W_o \tag{4}$$

Following multi-head attention, a fully connected feed-forward network (FFN) process is executed (Eq. (5)). This process involves the linear transformation of the output matrix from the multi-head attention step, followed by the application of the Rectified Linear Unit (ReLU) function. Parameters $W$ and $b$ are applied uniformly across positions, with variations introduced when transitioning across layers, hence referred to as position-wise transformation.

$$\text{FFN}(x) = \text{Max}(0, xW_1 + b_1)W_2 + b_2 \tag{5}$$

In our research, we introduce the DistilRoBERTa2GNN model, a novel architecture that combines the strengths of pre-trained language models with the advanced capabilities of GNNs for sentiment classification. This model is specifically designed to leverage the rich, contextual embeddings generated by DistilRoBERTa, a distilled version of the robust RoBERTa model renowned for its efficiency and effectiveness in processing natural language.

The first phase of our approach involves utilizing DistilRoBERTa as a feature extraction mechanism. DistilRoBERTa, a lighter yet highly potent version of its predecessor, provides

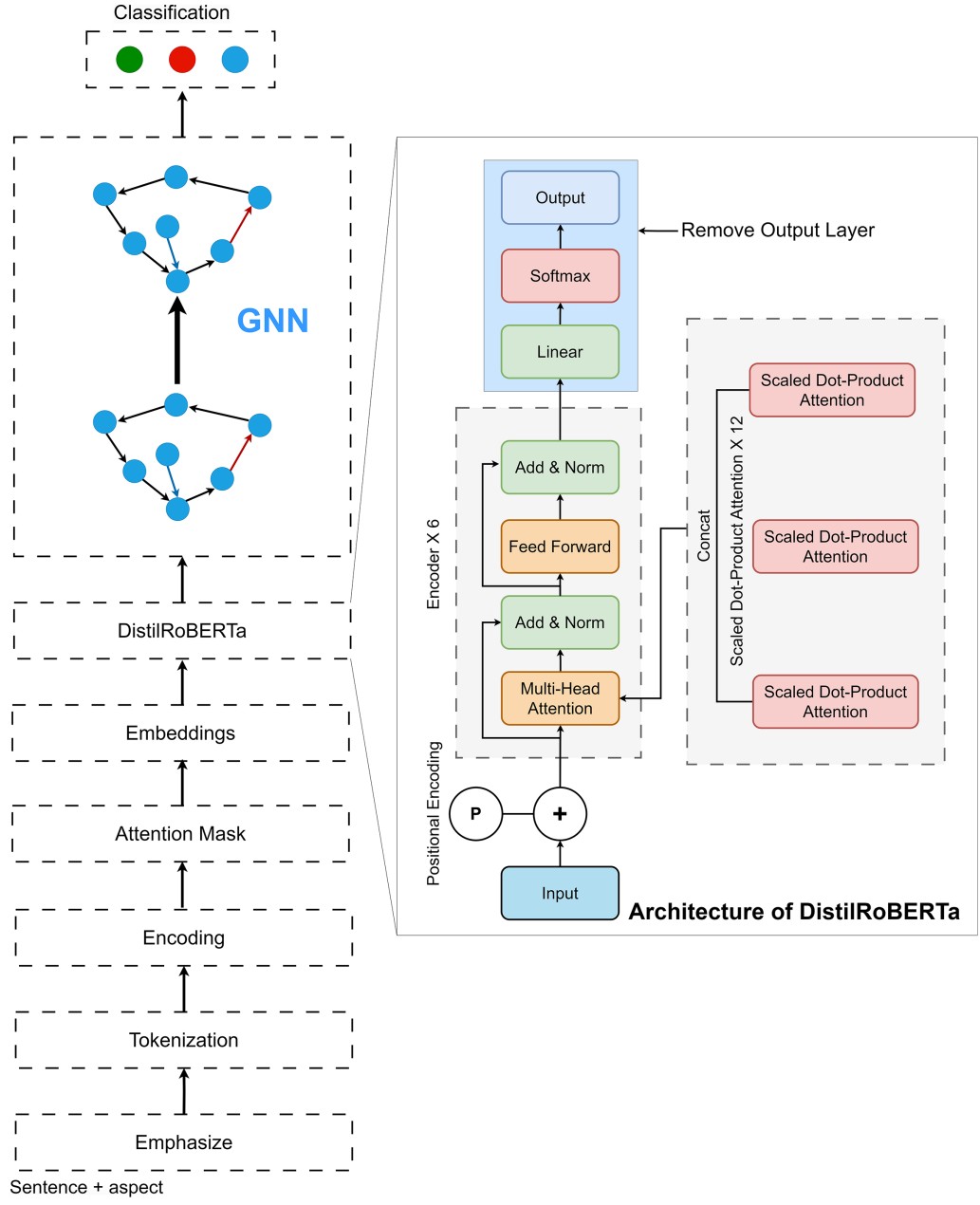

**Figure 2  Architecture of proposed DistilRoBERTa2GNN.**

deep contextualized embeddings that capture the nuances and complexities of language in a compact form. By processing textual data through DistilRoBERTa, we obtain high-quality feature representations that encapsulate the input text's semantic and syntactic characteristics. In doing so, we ensure that the intricate patterns and relationships within the text are preserved and made accessible for further analysis. Upon extracting these features, we strategically remove the output layer of DistilRoBERTa, shifting our focus

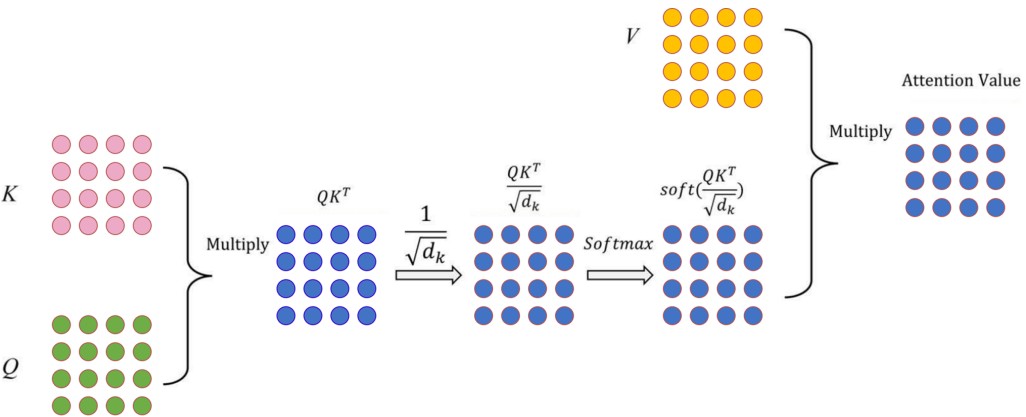

**Figure 3  Scaled dot-product attention technique.**

from the pre-trained model's original task to our specific objective of sentiment classification.

Following the feature extraction phase with DistilRoBERTa, our model transitions into the GNN component, where the extracted features are further processed and analyzed for sentiment classification. This seamless integration marks a pivotal step in our methodology, combining the nuanced textual understanding provided by DistilRoBERTa with the relational insight offered by GNNs. The GNN architecture is meticulously designed to handle the complexities of graph-structured data, beginning with the input layers that receive the contextual embeddings. These embeddings serve as the initial node features within the graph, representing individual pieces of text or entities within our dataset. To accommodate these features, our model employs reshaping layers, adjusting the dimensionality of the embeddings to fit the graph-based framework. This adjustment is crucial for aligning the data with the network's structure, ensuring that each node is represented in a manner that preserves its contextual integrity while facilitating efficient processing. Central to our GNN framework are the dropout layers, strategically placed to enhance the model's robustness against overfitting. By randomly omitting a subset of features during training, these layers encourage the model to develop more generalizable patterns, reducing its reliance on any single feature and improving its ability to perform sentiment classification across diverse datasets. The core of our GNN architecture is the mean aggregator layers, which play a vital role in synthesizing information across the graph. These layers aggregate features from the nodes' neighbours, effectively capturing the graph's local connectivity patterns and broader structural context. By averaging the features of neighbouring nodes, the mean aggregator layers ensure that each node's representation is informed by its immediate context, enriching the model's understanding of the relational dynamics at play.

Further processing and refinement of the aggregated features are achieved through additional reshaping and dropout layers. This hierarchical approach to feature processing streamlines the data and emphasizes the most salient features for sentiment classification. The culmination of this process is a transformation layer that prepares the features for the final classification task. This layer, often implemented through custom functions, adapts

the processed graph features for specific outputs, enabling our model to accurately predict sentiment based on the complex interplay of textual content and relational context. At its core, the DistilRoBERTa2GNN model stands as a trailblazing synthesis of deep linguistic insight and graph-based exploration. By capitalizing on the rich, contextual embeddings from DistilRoBERTa and implementing a refined GNN framework to probe the relational dynamics of the data, our model introduces a nuanced and all-encompassing strategy for sentiment classification. This innovative structure not only propels the domain of sentiment analysis forward but also underscores the value of combining diverse AI techniques to solve intricate challenges in language understanding. In our proposed approach, we standardized our training approach by setting the batch size to eight, employing the Adam optimizer for its efficiency in handling sparse gradients and adaptive learning rates, and setting the learning rate to 0.001. We conducted our experiments over 50 epochs, allowing for sufficient model training to capture the complexity of sentiment classification while avoiding overfitting. This configuration strikes a balance between computational efficiency and model performance, providing a robust framework for our analysis.

## Performance evaluation

In assessing the efficacy of the classification experiment, we employ the F1 score evaluation metrics (*Goutte & Gaussier, 2005*). The F1 score, depicted in Eq. (6), serves as a measure of the segmentation model's accuracy. It is calculated using the counts of true positives (TP), false positives (FP), and false negatives (FN). The F1 score represents the harmonic mean of precision and recall, as defined by Eqs. (7) and (8), respectively. Precision gauges the model's ability to accurately identify positive instances, while recall measures its capacity to capture all relevant instances.

$$\text{F1 score} = \frac{2 \times TP}{2 \times TP + FP + FN} \tag{6}$$

$$\text{Precision} = \frac{TP}{TP + FP} \tag{7}$$

$$\text{Recall} = \frac{TP}{TP + FN} \tag{8}$$

The F1 score proves to be an advantageous metric for simultaneously evaluating the precision and recall of the segmentation model.

## Computing environment

We conducted our ABSA research utilizing the comprehensive computational resources provided by Kaggle. This platform equipped us with a Tesla T4-16GB GPU and 32 GB of RAM, essential for the reproducibility of our experiments across various research endeavours. Kaggle's environment meets our computational demands and fosters a collaborative and accessible research landscape. This consistent computing framework is crucial, allowing us to focus on the innovative aspects of our study, ensuring that our

findings are reliable and facilitating their replication and expansion by the broader research community.

## RESULTS

In this section, we present the findings from our ABSA conducted on four publicly available benchmark datasets: Rest14, Rest15, Rest16-EN, and Rest16-ESP. Our methodology encompasses a series of data preprocessing steps designed to optimize the datasets for analysis, ensuring that the input data is in the most suitable format for our model. We introduce an innovative hybrid deep learning framework, DistilRoBERTa2GNN, which combines the powerful DistilRoBERTa model (a distilled version of the RoBERTa model, which itself is a robustly optimized BERT pretraining approach) with GNNs. This innovative strategy utilizes DistilRoBERTa's exceptional ability to understand contextual relationships within text, alongside GNN's proficiency in capturing and exploiting structural information inherent in data. Integrating these technologies facilitates more nuanced and accurate sentiment analysis at the aspect level.

### Baseline models

To evaluate the efficacy of our model, it is essential to conduct a comparative analysis with established baseline models across four distinct datasets. This approach is crucial for several reasons. First, it provides a clear benchmark, allowing us to position our model within the existing solutions landscape and understand its relative performance. By comparing against baselines, we can identify strengths and potential areas for improvement in our model, offering insights into how it adapts to different data types and challenges. Furthermore, this comparison underlines the model's innovation or superiority by highlighting its advancements over traditional or widely-used methods. Such an evaluation validates our model's effectiveness and robustness and contributes to the broader academic and practical knowledge, guiding future research and development efforts in the field.

1. ATAE-LSTM computes attention scores with respect to a specific aspect, enabling the model to focus on the most relevant parts of a sentence for sentiment analysis, thereby enhancing the precision of aspect-based sentiment prediction (*Wang et al., 2016*).
2. TNet-LF combines CNN and long short-term memory (LSTM) layers to extract relevant features from lexical representations, offering a robust framework for understanding the nuanced sentiment related to specific aspects (*Li et al., 2018*).
3. GCAE employs a gated CNN architecture, which selectively filters sentiment features relevant to the targeted aspect, optimizing the flow of information for precise sentiment analysis (*Xue & Li, 2018*).
4. RAM utilizes a recurrent attention-based memory network to identify sentiment features across long distances within the text, effectively capturing complex sentiment expressions related to aspects (*Chen et al., 2017*).

5. TD-LSTM employs dual LSTM networks that are target-dependent, separately processing the left and right contextual information related to an aspect, thus accurately capturing sentiment orientation (*Tang et al., 2015*).

6. AOA leverages an attention mechanism to highlight interactions between the sentence context and aspects, focusing on the most salient parts for sentiment prediction (*Huang, Ou & Carley, 2018*).

7. MemNet uses a multi-hop attention mechanism over a memory network to specifically concentrate on aspect words within sentences, enhancing ABSA (*Tang, Qin & Liu, 2016*).

8. MGAN differentiates between fine-grained and coarse-grained attention to better learn the interplay between aspect and context words, leading to a nuanced understanding of sentiments (*Fan, Feng & Zhao, 2018*).

9. IAN, through interactive attention networks, dynamically learns the relationship between context and aspect words, fostering a deeper understanding of sentiment influences (*Ma et al., 2017*).

10. ASCNN: This approach simplifies ASGCN by replacing the two GCN layers with two CNN layers (*Zhang, Li & Song, 2019*).

11. BiGCN introduces an innovative hierarchical structure that merges lexical and syntactical graphs. It utilizes a comprehensive word-level graph for illustrating occurrences of words together and a hierarchical approach for distinguishing diverse types of dependency links or connections between word pairs (*Zhang & Qian, 2020*).

12. SenticGCN innovates by amalgamating sentiment insights from SenticNet with graph neural networks, thereby refining the dependency graphs associated with sentences. This novel sentiment-augmented graph approach adeptly identifies the unique emotional characteristics of various aspects and thoroughly delineates the connections between specific aspects and the surrounding context (*Liang et al., 2022*).

13. DGEDT extracts both flat and graph-based representations through a dual-transformer network enhanced by a dependency graph, capturing complex syntactic and semantic relationships (*Tang et al., 2020*).

14. ASGCN-DT builds a directional graph based on sentence dependency trees, utilizing a GCN to mine syntactical patterns and word dependencies for sentiment analysis (*Zhang, Li & Song, 2019*).

15. R-GAT advances sentiment prediction by encoding aspect-based tree structures using a relational graph attention network, effectively capturing relational nuances (*Wang et al., 2020*).

16. ASGCN-DG is akin to ASGCN-DT but utilizes an unidirectional adjacency matrix in its graph, simplifying the representation while maintaining effectiveness in extracting sentiment-related features (*Zhang, Li & Song, 2019*).

17. IA-HiNET enhances aspect-based sentiment analysis by combining a GCN with a Bi-LSTM framework to capture interdependent features between aspect words effectively. Utilizing part-of-speech and positional information as prior knowledge, along with an

information interaction mechanism, IA-HiNET refines and fuses sentence representations, improving the precision of sentiment predictions for specific aspects (*Gu, Zhao & Li, 2023*).

18. EK-GCN enhances aspect-based sentiment analysis by integrating external knowledge, such as sentiment lexicons and part-of-speech information, into a GCN. This model constructs sentiment score matrices and part-of-speech matrices to enrich the representation of words in sentences, thereby improving the accuracy of sentiment predictions for specific aspects across various datasets (*Gu et al., 2023*).

19. SS-GCN enhances aspect-based sentiment analysis by integrating syntactic and semantic information within a GCN framework to model relationships between words in a sentence effectively. The model leverages a syntactic weighted matrix and semantic graphs, enhancing text representation by focusing on both the structure and meaning of words relative to aspects. This dual approach allows SS-GCN to achieve a nuanced understanding of textual sentiment, significantly improving the accuracy of sentiment predictions across multiple datasets (*Chen, Fan & Wang, 2024*).

20. The MLGCN-DL model utilizes a dependency-based location-aware transformation function to enhance aspect-based sentiment analysis. By merging aspect entities into a single token and applying this novel location-aware function, MLGCN-DL improves the model's attention to opinion words relevant to the aspects (*Jiang, Xu & Liu, 2023*).

21. The MLGCN-SE model employs a Squeeze-and-Excitation Attention-based (SE-attention) location-aware transformation function to advance aspect-based sentiment analysis. This model integrates merged aspect entities and the SE-attention mechanism to focus on semantic representations of opinion words related to aspects (*Jiang, Xu & Liu, 2023*).

22. The ACS-DISTILL-M model combines aspect code-switching with multi-teacher distillation for cross-lingual ABSA. By training on various bilingual data and distilling from multiple teachers, it captures diverse language-specific and task-specific knowledge, enhancing cross-lingual alignment and achieving state-of-the-art performance (*Zhang et al., 2021*).

23. The TrAsp model enhances aspect-based sentiment analysis by using transformer-based embeddings and a multi-layer architecture that includes a linear layer, dropout, and a final linear layer for classification.This model excels in capturing nuanced language patterns, outperforming baseline methods in both exact and relaxed evaluation settings (*Szołomicka & Kocon, 2022*).

## Performance analysis and comparison with baseline models

### Performance analysis of our model

The efficacy of our model is evaluated across these datasets, and the results are comprehensively detailed below. The experimental results demonstrate that the fusion of DistilRoBERTa with GNNs marks a significant advancement in the field of sentiment analysis, promising to enhance both the precision and depth of insights derived from text

**Table 3 Evaluation metrics using proposed model.**

| Datasets | F1 score | Precision | Recall |
|---|---|---|---|
| Rest14 | 77.98% | 78.12% | 79.41% |
| Rest15 | 76.86% | 80.70% | 79.37% |
| Rest16-EN | 84.96% | 82.77% | 87.28% |
| Rest16-ESP | 74.87% | 73.11% | 76.80% |

data. The evaluation metrics used to assess the model's efficacy include the F1 score, precision, and recall, providing a comprehensive view of its performance across various scenarios. Table 3 illustrates the performance of our proposed approach with four datasets.

For the Rest14 dataset, the model achieved an F1 score of 77.98%, demonstrating robust performance. This metric, indicative of the model's balanced precision and recall, reflects its effectiveness in harmonizing the true positive rate with the precision of its predictions. Specifically, a precision of 78.12% suggests that the model accurately identifies sentiments as positive or negative. A recall rate of 79.41% indicates the model's efficiency in capturing relevant instances of sentiment within the dataset.

The analysis of the Rest15 dataset revealed an F1 score of 76.86%, showcasing a slight improvement in the balance between precision and recall compared to the Rest14 dataset. With a precision of 80.70% and a notably higher Recall of 79.37%, this increase in recall suggests an enhancement in the model's ability to identify all relevant instances of sentiment, albeit at a slight expense of precision.

The Rest16-EN dataset highlighted the model's superior performance, with an F1 score of 84.96%. This significant improvement reflects the model's enhanced capability to identify and classify sentimental expressions accurately. The precision and recall were measured at 82.77% and 87.28%, respectively, highlighting the model's exceptional accuracy and completeness in capturing sentiment expressions within this dataset.

For the Rest16-ESP dataset, a Spanish-language dataset, the DistilRoBERTa2GNN model achieved an F1 score of 74.87%, precision of 73.11%, and recall of 76.80%. These metrics underscore the model's balanced performance in identifying and classifying sentiments accurately. The proposed model demonstrates a notable improvement in sentiment analysis tasks across various datasets, showcasing its robustness and effectiveness in handling complex sentiment expressions.

The experimental results from the evaluation of the proposed model across four distinct datasets demonstrate its robust and versatile performance in sentiment analysis tasks, encompassing both English and Spanish datasets. Characterized by high F1 scores and closely matched precision and recall rates, these results indicate the model's efficacy in accurately and comprehensively identifying sentiments across various contexts. This study contributes valuable insights into the model's applicability and effectiveness in sentiment analysis, showcasing its potential for wide-ranging applications in natural language processing and related fields.

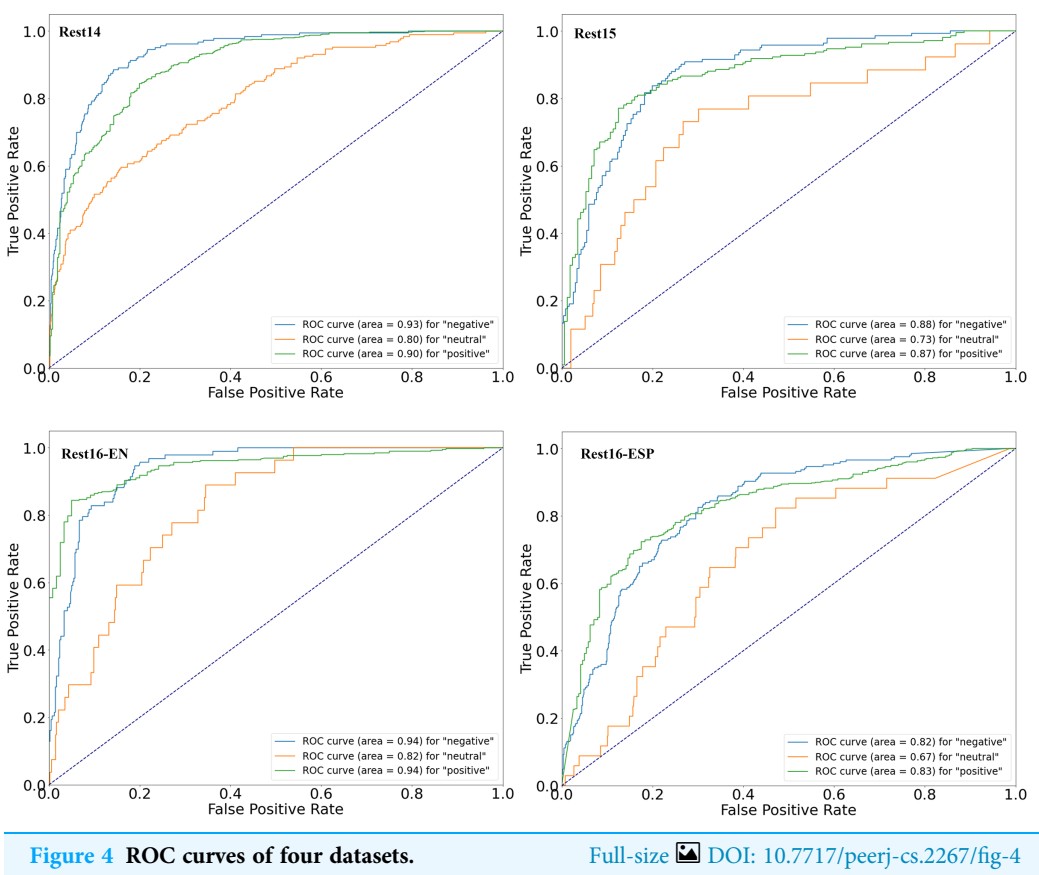

**Figure 4** ROC curves of four datasets.           

The four figures (in Fig. 4) illustrate the performance of a proposed sentiment analysis model across four datasets using receiver operating characteristic (ROC) curves to assess its ability to classify different sentiment categories: "negative," "neutral," and "positive." For Rest14, the model showed high accuracy in classifying "negative" (AUC: 0.93) and "positive" (AUC: 0.90) sentiments, with moderate accuracy for "neutral" sentiments (AUC: 0.80). In the Rest15 dataset, the model also performed well, particularly in identifying "negative" (AUC: 0.88) and "positive" (AUC: 0.87) sentiments, though it was less effective for "neutral" sentiments (AUC: 0.73). The Rest16-EN dataset revealed superior performance, with the model achieving an AUC of 0.94 for "negative" and 0.94 for "positive" sentiments, indicating a strong discriminative power. In contrast "neutral" sentiments were classified with good accuracy (AUC: 0.82). The current ROC curve demonstrates the model's capacity to differentiate between sentiment categories with an AUC of 0.82 for "negative", 0.67 for "neutral", and 0.83 for "positive", respectively, across the evaluated dataset. These curves significantly surpass the baseline performance of a random classifier, highlighted by a diagonal dashed line, underscoring the model's effectiveness in sentiment analysis.

## Performance comparison with baseline models
In this section, we present a detailed comparative analysis of the proposed DistilRoBERTa2GNN model against established baseline models across four distinct

**Table 4 Performance comparison of our proposed model with different baseline models used in ABSA studies.**

| Models | F1 score | | | |
| --- | --- | --- | --- | --- |
| | Rest14 | Rest15 | Rest16-EN | Rest16-ESP |
| ATAE-LSTM | 67.02% | 60.53% | 61.71% | – |
| TNet-LF | 71.03% | 59.47% | 70.43% | – |
| GCAE | 62.45% | 56.03% | 62.69% | – |
| RAM | 70.80% | 60.57% | 62.14% | – |
| TD-LSTM | 66.73% | 58.70% | 54.21% | – |
| AOA | 70.42% | 57.02% | 66.21% | – |
| MemNet | 69.64% | 58.28% | 65.99% | – |
| MGAN | 71.94% | 57.26% | 62.29% | – |
| IAN | 70.09% | 52.65% | 55.21% | – |
| ASCNN | 73.10% | 58.90% | 64.56% | – |
| BiGCN | 73.48% | 64.79% | 70.84% | – |
| SenticGCN | 75.38% | 67.32% | 75.91% | – |
| DGEDT | 75.10% | 65.90% | 73.80% | – |
| ASGCN-DG | 72.02% | 61.89% | 67.48% | – |
| R-GAT | 76.08% | 64.17% | 70.89% | – |
| ASGCN-DT | 72.19% | 60.78% | 66.64% | – |
| IA-HiNET | 75.85% | – | 72.68% | – |
| SS-GCN | 74.26% | 66.16% | 74.28% | – |
| EK-GCN | 74.93% | – | 69.32% | – |
| MLGCN-DL | 77.61% | 68.08% | 73.77% | – |
| MLGCN-SE | 71.89% | 67.83% | 76.05% | – |
| IXA-pipes | 84.11% | 70.90% | 73.51% | 69.92% |
| ACS-DISTILL-M (mBERT) | – | – | – | 62.91% |
| ACS-DISTILL-M (XLM-R) | – | – | – | 70.38% |
| TrAsp (LaBSE) | – | – | 70.30 | 69.19 |
| TrAsp (XLM-R) | – | – | 74.93 | 72.97 |

datasets: Rest14, Rest15, Rest16-EN, and Rest16-ESP. The primary metric for this evaluation is the F1 score, which is a balanced measure of a model's precision and recall capabilities. The comparative results, as depicted in Table 4, underscore the superior performance of the DistilRoBERTa2GNN model across all evaluated datasets.

The DistilRoBERTa2GNN model achieved an F1 score of 77.98% on the Rest14 dataset, surpassing the next best model, R-GAT, which recorded a score of 76.08%. This improvement of nearly 1.90% is statistically significant and highlights the advanced capabilities of our model in processing complex linguistic structures and sentiment nuances inherent in the dataset.

In the Rest15 dataset, our model demonstrated remarkable robustness, achieving an F1 score of 76.87%. This score significantly exceeds that of the second-best performing model, SenticGCN, which achieved a score of 67.32%. The margin of 9.55% between the two models illustrates the superior contextual understanding and analytical depth of the DistilRoBERTa2GNN, making it highly effective for tasks requiring nuanced sentiment analysis.

For the Rest16-EN dataset, the DistilRoBERTa2GNN model provide highest performance, achieving an F1 score of 84.96%, compared to the 75.91% scored by SenticGCN. The difference of 9.05% underlines the model's consistency and adaptability across linguistically and contextually diverse datasets, indicating a robust generalization capability.

The model's effectiveness is further validated by its performance on the Rest16-ESP dataset, where it achieved an F1 score of 74.87%. This score surpasses that of IXA-pipes, which achieved 69.92%, showcasing the DistilRoBERTa2GNN model's proficiency in multilingual sentiment analysis. This capability is critical for deploying sentiment analysis solutions in a globalized market.

The consistent outperformance of the DistilRoBERTa2GNN model across diverse datasets is attributed to its innovative architecture, which effectively integrates distilled transformer representations with graph neural network techniques. This integration allows the model to capture and analyze complex sentiment expressions more accurately and efficiently than traditional models. The findings from this study not only reinforce the effectiveness of combining these advanced technologies but also open up new avenues for further research into scalable and adaptable sentiment analysis tools.

## CONCLUSIONS

This study developed a cutting-edge approach to ABSA by introducing a novel hybrid architecture, DistilRoBERTa2GNN. This method capitalizes on the synergistic potential of the DistilRoBERTa pre-trained model and a GNN to significantly enhance the accuracy and efficiency of sentiment analysis. Within this framework, the DistilRoBERTa model extracts informative features from datasets, and then the GNN efficiently categorizes the sentiment based on these features. To further optimize the model's effectiveness, a four-phase data preprocessing strategy is applied to the raw data, aimed at removing unnecessary information and thus improving performance. The robustness of our proposed model has been validated against four publicly available benchmark datasets. Our experimental analysis conclusively demonstrates that our ABSA approach significantly outperforms existing methods. Additionally, when compared against various baseline models, our methodology consistently exhibited superior performance. In future, we aim to explore the integration of additional contextual layers within the DistilRoBERTa2GNN framework to refine its understanding of nuanced sentiments further. We also plan to extend our methodology to multilingual datasets, broadening the applicability of our approach to global sentiment analysis challenges.

### Funding

This work was funded by the Researchers Supporting Project number (RSPD2024R532), King Saud University, Riyadh, Saudi Arabia. The funders had no role in study design, data collection and analysis, decision to publish, or preparation of the manuscript.

### Grant Disclosures

The following grant information was disclosed by the authors:
Researchers Supporting Project number: RSPD2024R532.
King Saud University.

### Competing Interests

The authors declare that they have no competing interests.

### Author Contributions

- Aseel Alhadlaq conceived and designed the experiments, performed the experiments, analyzed the data, performed the computation work, prepared figures and/or tables, authored or reviewed drafts of the article, and approved the final draft.
- Alaa Altheneyan conceived and designed the experiments, performed the experiments, analyzed the data, performed the computation work, prepared figures and/or tables, authored or reviewed drafts of the article, and approved the final draft.

### Data Availability

The Rest 14 dataset is available at QCRI: https://alt.qcri.org/semeval2014/task4/index.php?id=data-and-tools.

The Rest 15 dataset is available at QCRI: https://alt.qcri.org/semeval2015/task12/index.php?id=data-and-tools.

The Rest 16-EN dataset is available at QCRI: https://alt.qcri.org/semeval2016/task5/index.php?id=data-and-tools.

The Rest 16-ESP dataset is available at QCRI: https://alt.qcri.org/semeval2016/task5/index.php?id=data-and-tools.

Code is available at GitHub: https://github.com/sawikot/Sentiment-Analysis.git.

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
