# Peer review of "Distilroberta2gnn: a new hybrid deep learning approach for aspect-based sentiment analysis"

_PeerJ Computer Science, doi:10.7717/peerj-cs.2267_

## Round 0.1 · original submission · Minor Revisions

Thank you for submitting your paper to PeerJ computer science. The authors suggest to assign a minor revision to this paper but they are consistent with their feedback.

Reviewer 1 ·

Basic reporting

This study introduces a new ABSA (Aspect-Based Sentiment Analysis) approach called DistilRoBERTa2GNN, which suggests DistilRoBERTa2GNN model. Model employes DistilRoBERTa pretrained model and a Graph Neural Network (GNN). Before the DistilRoBERTa2GNN are applied, the pre-processing is required. This preprocess is composed of four-phase data preprocessing (emphasize tokenization, encoding, attention mask) in roder to remove unnecessary information. Dataset are employed Rest 14, Rest 15 and Rest 16, which are reviews on restaurant.

Experimental design

An ablation study is imperative to showcase the comparative advantage of employing the combination of preprocessing and DistilRoBERTa independently over alternative models integrating Graph Neural Networks (GNNs). Recent scholarly articles either employ solely one of the two models to substantiate the indispensability of their fusion or undertake an ablation study to determine the optimal configurations.

Validity of the findings

The baseline models featured in the paper lack current relevance. They primarily compare against models developed between 2015 and 2022. It is necessary to identify the differences between recent papers related to ASBA (Aspect-Based Sentiment Analysis) and benchmark them against them from 2023 onwards. Here are recently published papers.
- Syntactic and Semantic Aware Graph Convolutional Network for Aspect-Based Sentiment Analysis, IEEE Access, 2024
- Aspect-level sentiment classification via location enhanced aspect-merged graph convolutional networks, Journal of Supercomputing, 2023

Reviewer 2 ·

Basic reporting

Good

Experimental design

to improve

Validity of the findings

The article is about a Distilroberta2gnn a new hybrid deep learning approach for aspect-based sentiment analysis. This paper is not acceptable in its current form but has merit. After the corrections presented by the authors, it is suitable for publication in the journal.
1. Please compare the performance of your proposed approach with other recently published approaches to ensure a fair comparison.
2. The authors will provide more experimental results to prove the superiority of the proposed approach as compared to the existing methods, especially on the multilingual data set. This would improve the robustness of the proposed model.

---

## Round 0.2 · accepted · Accept

Thank you for submitting your revised version to PeerJ journal. The reviewers suggest to accept the paper.

Reviewer 1 ·

Basic reporting

This study introduces the DistilRoBERTa2GNN model, a hybrid approach combining DistilRoBERTa’s feature extraction with Graph Neural Networks (GNN) for dynamic sentiment classification. In the revision, we added REST 16 ESP data and performed additional experimental comparisons to compare performance differences with other ASBA models.
The authors have revised the manuscript based on the provided comments. With corrections to English notation and improvements to the overall flow, I believe it is suitable for publication.

Experimental design

One of the main topics of the journal Peerj Computer Science is sentimental analysis. This manuscript can analyze human sentiments in the text by utilizing natural language processing and a graph model that has been used recently. Therefore, this manuscript can refer to recent technology trends.

Validity of the findings

no comment

Reviewer 2 ·

Basic reporting

good

Experimental design

Good

Validity of the findings

After the corrections provided by the authors, it is suitable for publication in the journal.